# Gold Nanoparticles-Functionalized Ultrathin Graphitic Carbon Nitride Nanosheets for Boosting Solar Hydrogen Production: The Role of Plasmon-Induced Interfacial Electric Fields

**DOI:** 10.3390/molecules30163406

**Published:** 2025-08-18

**Authors:** Haidong Yu, Ziqi Wei, Qiyue Gao, Ping Qu, Rui Wang, Xuehui Luo, Xiao Sun, Dong Li, Xiao Zhang, Jiufen Liu, Liang Feng

**Affiliations:** 1Langfang Natural Resources Comprehensive Survey Center, China Geological Survey, Langfang 065000, China; yuhaidong1992@163.com (H.Y.); 13115333803@163.com (Z.W.); gqy7770119@163.com (Q.G.); wrui@mail.cgs.gov.cn (R.W.); ls19760811@163.com (X.L.); sunxiao@mail.cgs.gov.cn (X.S.); 17852033981@163.com (D.L.); xiao.zhang@cug.edu.cn (X.Z.); 2Center for Geophysical Survey, China Geological Survey, Langfang 065000, China; quping_cgs@163.com; 3Natural Resources Comprehensive Survey Command Center, China Geological Survey, Beijing 100055, China

**Keywords:** Au nanoparticles, LSPR, hydrogen evolution, interfacial electric field

## Abstract

The design of photocatalysts capable of generating localized surface plasmon resonance (LSPR) effects represents a promising strategy for enhancing photocatalytic activity. However, the mechanistic role of plasmonic nanoparticles-induced interfacial electric fields in driving photocatalytic processes remains poorly understood. To produce a Schottky junction, varying amounts of Au nanoparticles widely utilized to broaden the light absorption were loaded onto ultrathin carbon nitride sheets (Au/UCN). The Au/UCN-20 Schottky junction exhibits exceptional photocatalytic activity, achieving a hydrogen evolution rate (14.2 mmol·g^−1^ over a 4 h period) while maintaining robust stability through five consecutive photocatalytic cycles. The LSPR activity of Au nanoparticles are responsible for the broadened light absorption spectrum of Au/UCN nanocomposites. The interfacial electric field generated at the Au /UCN heterojunction is proposed to enhance charge-transfer efficiency through Schottky barrier penetration of photocarriers, mediated by electric field-driven carrier migration, according to surface potential and finite-difference time-domain (FDTD). These findings uncover a previously obscured photocatalytic mechanism driven by LSPR-induced interfacial electric fields, pioneering a quantum-dot-directed strategy to precisely engineer charge dynamics in advanced photocatalysts via targeted manipulation of nanoscale electric field effects.

## 1. Introduction

Clean and sustainable energy has become a critical and urgent requirement for minimizing the excessive use of limited fossil fuels and addressing major environmental challenges [1]. Solar energy, being an inexhaustible clean energy source that fuels all life on Earth, is thought to be the most exploitable [2]. Photocatalytic water splitting is regarded as a viable and environmentally friendly method for converting solar energy into pure hydrogen [3,4]. With a utilization rate of less than 50% of the solar energy, the majority of photocatalysts used for this conversion are semiconductor materials with broad band gaps (TiO_2_, ZnO, Fe_2_O_3_, and WO_3_), which limits it to the visible or ultraviolet area [5,6,7,8]. Therefore, it is important to figure out how to make sure that near-infrared (NIR) photons are used efficiently in solar energy and to further implement full-spectrum photocatalytic water splitting.

Currently, numerous kinds of photocatalysts have been produced, such as metal oxides, chalcogenides, carbon nitride, and other metal–organic framework compounds [9,10,11,12]. It was reported that *g*-C_3_N_4_ has an ultrathin nanosheet structure in two dimensions. Owing to its exceptional structural and optoelectronic properties—including enhanced surface area, pore volume, robust stability, a tailored bandgap structure, and a distinctive two-dimensional layered architecture—this material exhibits remarkable photoelectric response and photocatalytic efficiency. These attributes collectively contribute to its widespread application in solar-driven water splitting systems [13,14]. However, the recombination of photoexcited holes and electrons as well as the limited number of active sites are the two primary problems restricting the usage of *g*-C_3_N_4_ as a photocatalyst. A variety of strategies have been investigated to overcome the *g*-C_3_N_4_ bottleneck issues mentioned above [15]. These strategies include co-catalyst loading, surface modifications, and heterojunction construction [16,17].

Decorating ultrathin *g*-C_3_N_4_ sheets with plasmonic metal particles (Au, Ag) can improve photocatalytic performance by acting as a charge sink and strong LSPR effect [18,19,20]. The surface electrical structure of the support is significantly influenced by the size and shape of particles [21]. In catalytic reactions, the low utilization efficiency of metal atoms often results in poor activity or performance for plasmonic metal co-catalysts with larger particle sizes. As the metal particle size decreases, the number of surface low-coordination atoms progressively increases, leading to higher exposure ratios. Concurrently, the interfacial contact area between the metal co-catalyst and the catalyst expands. This synergistic effect not only enhances the atomic utilization efficiency of the metal co-catalyst but also induces significant modifications in both the structural properties and active site distribution of the catalytic material [22]. Since Au nanoparticles may induce energy transfer, facilitate hot electron injection, and magnify electron scattering, their optimal energy level and LSPR effect have drawn attention from the general public [23,24,25]. The immobilization of gold nanoparticles on ultrathin *g*-C_3_N_4_ photocatalysts represents an effective strategy for performance enhancement, originating from the distinctive properties of Au and their ability to optimize the catalyst’s electronic structure. Investigating the photocatalytic reaction kinetics and the interfacial electric field induced by LSPR constitutes another crucial aspect of this research.

Herein, we report the in situ synthesis of plasmonic Au nanoparticles anchored on *g*-C_3_N_4_ nanosheets, constructing the Au/UCN heterojunction photocatalyst for efficient broad-spectrum solar-driven water splitting. Under UV-range excitation, the Au/UCN system exhibits enhanced photocatalytic activity primarily attributed to the intrinsic bandgap transition of *g*-C_3_N_4_. Meanwhile, the LSPR effect of Au quantum dots functions as a Vis-NIR light harvester, generating hot charge carriers that inject into *g*-C_3_N_4_ to drive visible-light photocatalytic processes. In order to provide unique and beneficial entire-solar-spectrum photocatalytic water splitting, the results showed that the LSPR of Au effectively extended the photocatalytic activity of UCN-based photocatalyst to NIR light. To elucidate the fundamental mechanism underlying the high catalytic activity, this study adopts an integrated research strategy combining theoretical calculations with experimental verification. By synergizing surface potential distribution analysis with finite-difference time-domain (FDTD) simulations, we reveal the physical mechanism by which the synergistic interaction between localized surface plasmon resonance (LSPR) effects and in-plane electric fields regulates photogenerated carrier migration. These systematic findings provide crucial theoretical guidance for the structural design of quantum-dot-based composite photocatalysts.

## 2. Results and Discussion

### 2.1. Structural Characterization

X-ray diffraction (XRD) analysis was systematically employed to characterize the crystal structures of both pristine UCN nanomaterials and the Au/UCN nanocomposite system formed by combining with gold nanoparticles. The XRD patterns (Figure 1a) revealed distinct characteristic diffraction peaks at 2θ = 12.7° and 27.0° for the unmodified UCN sample, which were indexed to the (100) and (002) crystallographic planes of the hexagonal phase, respectively [26]. Studies indicate that these characteristic diffraction peaks originate from the unique layered structural arrangement and molecular stacking configuration of UCN materials [27]. In the XRD profile of the Au/UCN composite, in addition to the characteristic peaks of UCN, new diffraction signals were detected at 38.1°, 44.5°, 64.4°, and 77.5°. These additional peaks show perfect correspondence with the (111), (200), (220), and (311) planes of face-centered cubic gold nanocrystals [28]. Notably, with increasing Au content, the intensity of UCN characteristic peaks at 12.7° and 27.0° gradually decreased, accompanied by peak broadening. This phenomenon can be attributed to the incorporation of Au into the (002) crystal plane of UCN, leading to structural distortion and fragmentation. Furthermore, as illustrated in Figure 1b, the characteristic (002) peak of UCN undergoes a slight shift, suggesting strong interfacial interactions between Au nanoparticles and UCN, as well as the weakening of interlayer van der Waals forces in UCN upon modification with Au nanoparticles [29].

X-ray photoelectron spectroscopy (XPS) analysis was conducted to investigate the elemental composition and chemical states of the Au/UCN composite (Figure 2). The survey spectrum (Figure 2a) exhibits characteristic peaks corresponding to Au, C, and N elements, confirming the coexistence of metallic gold and *g*-C_3_N_4_ in the composite. High-resolution XPS spectra provide further insights into the chemical bonding characteristics; the deconvoluted C 1s spectrum (Figure 2b) reveals two distinct components at 284.6 eV and 287.9 eV [30]. These binding energies are characteristic of graphitic carbon (C-C) and sp^2^-hybridized carbon atoms in the heterocyclic N-C=N structure of *g*-C_3_N_4_, respectively. Similarly, the N 1s spectrum (Figure 2c) displays two resolved peaks at 398.5 eV and 400.0 eV, corresponding to sp^2^-bonded nitrogen in C-N=C coordination and tertiary nitrogen atoms in N-(C)_3_ groups [31]. Notably, the Au 4f spectrum (Figure 2d) exhibits spin–orbit split peaks at 83.0 eV (Au 4f_7/2_) and 86.7 eV (Au 4f_5/2_) with a splitting energy of 3.7 eV, which is characteristic of metallic gold (Au^0^) [32]. Compared to the standard Au^0^ reference values (84.0 eV for Au 4f_7/2_ and 87.7 eV for Au 4f_5/2_), the observed negative shifts in binding energies suggest strong interfacial interactions and electron transfer between Au nanoparticles and the UCN substrate [33]. Complementary FT-IR analysis (Appendix A) confirms the preservation of *g*-C_3_N_4_’s structural integrity after Au nanoparticle decoration. Both pristine *g*-C_3_N_4_ and Au/UCN composites exhibit characteristic vibrational modes: the sharp peak at 810 cm^−1^ corresponds to out-of-plane bending vibrations of triazine rings; multiple peaks between 1200 and 1700 cm^−1^ arise from stretching vibrations of C-N heterocycles; a weak absorption at 2180 cm^−1^ indicates the presence of terminal cyano groups (-C≡N); and broad bands in the 3100–3500 cm^−1^ region are attributed to N-H stretching vibrations. The FT-IR spectra show no significant alterations in peak positions or relative intensities after Au incorporation, demonstrating that the Au does not induce detectable changes in the crystal structure or chemical bonding configuration of the *g*-C_3_N_4_ matrix [34].

The morphology and nanostructural characteristics of the Au/UCN were systematically investigated through TEM analysis (Figure 3). As shown in Figure 3a, the Au nanoparticles demonstrate uniform anchoring on the UCN substrate, with intimate interfacial contact observed between the Au nanostructures and the *g*-C_3_N_4_ matrix—a crucial feature for facilitating efficient interfacial charge transfer. Elemental mapping analysis confirms the homogeneous distribution of Au species across the carbon nitride support. High-resolution TEM (HRTEM) examination (Figure 3b) reveals well-defined lattice fringes with a measured spacing of 0.235 nm, corresponding to the (111) crystallographic plane of face-centered cubic Au [23]. Notably, while conventional Au nanoparticles typically exhibit average diameters of ~19 nm [30], the synthesized composite shows significantly reduced Au particle sizes of approximately 10 nm. This marked size reduction suggests that the *g*-C_3_N_4_ substrate exerts a pronounced size-confinement effect during nanoparticle nucleation and growth, likely through its rich nitrogen-containing functional groups that act as anchoring sites.

The combination of morphological observations, crystallographic analysis, and elemental distribution profiles provides conclusive evidence for the quantum dot nature of the Au species in the composite. These TEM findings, when correlated with complementary HRTEM, XRD, and XPS data presented in earlier sections, collectively demonstrate the successful fabrication of the Au/UCN heterostructure with well-defined interfacial characteristics.

### 2.2. Photoelectrochemical Performance

Employing transmissive photocurrent detection technique, this investigation systematically explores the photoinduced charge carrier dynamics in upconversion nanoparticles (UCNs) and their hybrid systems with gold nanoparticles (Au/UCNs), with particular emphasis on charge-transfer kinetics and separation efficiency. Experimental results (Figure 4a) revealed that both nanostructures exhibited pronounced photoresponsive behavior under visible-light excitation. Upon light illumination, the system promptly generated detectable photocurrent signals, while immediate current attenuation was observed following illumination cessation. This distinct temporal response pattern conclusively demonstrates the material’s superior capabilities in photoinduced charge separation and transfer. The photocurrent density progression followed this hierarchical order: UCN < Au/UCN-5 < Au/UCN-10 < Au/UCN-15 < Au/UCN-30 < Au/UCN-20. Complementary electrochemical impedance spectroscopy (EIS) analysis (Figure 4b) revealed interfacial charge-transfer characteristics through Nyquist plot evaluation [35]. The arc radius progression was inversely correlated with the photocurrent performance in the following order: UCN > Au/UCN-5 > Au/UCN-10 > Au/UCN-15 > Au/UCN-30 > Au/UCN-20. The minimal semicircular radius observed for Au/UCN-20 signifies two critical advantages: (1) significantly reduced charge-transfer resistance, and (2) enhanced interfacial charge migration kinetics. This synergistic enhancement explains the superior photoelectrochemical performance of the Au/UCN-20 heterostructure. Steady-state photoluminescence (PL) spectroscopy was employed to investigate recombination dynamics of photogenerated electron–hole pairs in the semiconductor systems (Figure 4c). Pristine UCN exhibited maximum emission intensity at 460–465 nm, which is characteristic of intrinsic band-edge recombination. Notably, Au nanoparticles incorporation induced a bathochromic shift in the PL peak to 475–480 nm, a phenomenon attributed to the hybridization of Au’s 4f orbitals with the host matrix. This electronic interaction effectively modifies the band structure by filling mid-gap states, thereby reducing the energy disparity between π* antibonding and lone-pair (LP) orbitals [36]. To elucidate the mechanism of Au nanoparticles in suppressing electron–hole recombination, time-resolved photoluminescence (TRPL) spectroscopy was employed to probe carrier dynamics. Comparative analysis revealed a substantial reduction in the average electron lifetime from 3.14 ns for pristine UCN to 0.41 ns for the Au/UCN-20 composite (Figure 4d). This pronounced lifetime quenching unambiguously demonstrates that Au nanoparticles act as efficient electron sinks, facilitating rapid electron extraction from the UCN host matrix to its surface. The accelerated charge migration kinetics directly correlate with suppressed carrier recombination, thereby enhancing the availability of photogenerated electrons for surface-mediated redox processes.

UV-vis diffuse reflectance spectroscopy (DRS) was systematically employed to evaluate the light-harvesting properties of pristine UCN and its Au/UCN-20 composite (Figure 5a). Both materials exhibited a fundamental absorption edge near 450 nm, characteristic of the host semiconductor. Notably, the Au/UCN-20 heterostructure demonstrated markedly enhanced absorption in the 700–800 nm range, a phenomenon ascribed to the localized surface plasmon resonance (LSPR) of metallic Au nanoparticles [37]. Furthermore, bandgap engineering was quantitatively analyzed using Tauc plot methodology derived from the Kubelka–Munk transformation, (αhν) = A(hν − Eg)^n/2^ [38], where the Eg values of UCN and Au/UCN-20 are 2.71 and 2.42 eV, respectively (Figure 5b), demonstrating two critical effects: 1. band structure modulation: Au incorporation introduces intragap states that narrow the effective bandgap; 2. LSPR synergy: plasmonic excitation of Au enables sub-bandgap photon utilization. This dual mechanism synergistically enhances light absorption cross-sections while promoting hot electron injection, establishing a photophysical foundation for the observed photocatalytic enhancement.

### 2.3. Photocatalytic Hydrogen Evolution

Photocatalytic hydrogen evolution experiments were conducted under full-spectrum simulated solar irradiation (UV-vis-NIR) and wavelengths longer than 420 nm (vis-NIR) and 700 nm (NIR) light irradiation, using triethanolamine (TEOA) as a hole scavenger to suppress charge recombination. Blank control experiments confirmed the essentiality of both photocatalyst presence and photon activation, as no measurable hydrogen generation occurred in their absence. The activity hierarchy of the synthesized materials followed this descending order: Au/UCN-20 > Au/UCN-30 > Au/UCN-15 > Au/UCN-10 > Au/UCN-5 > UCN. The Au/UCN-20 composite demonstrated exceptional performance, achieving a cumulative hydrogen yield of 14.2 mmol·g^−1^ over 4 h with linear production kinetics under UV-vis-NIR (Figure 6a). The detailed hydrogen evolution rates of UCN and Au/UCN photocatalysts were shown in Appendix A. In order to investigate the absorption and utilization efficiency of catalysts for light in different wavelength ranges, we conducted comparative experiments on UCN and Au NPs/UCN-20, separately. From Appendix A, the amount of H_2_ generated from Au NPs/UCN-20 sample increased with the time and reached 19.4 mmol·g^−1^ under solar irradiation (UV-vis-NIR) for 5 h. Based on Appendix A, we obtained the evolution of H_2_ under irradiation from UV, visible, and NIR light. Under NIR light irradiation, the H_2_ evolution is at its maximum compared to production capacities under UV and visible-light illumination (Figure 6b), consistent with the full-spectrum absorption properties of UCN and Au NPs /UCN-20 (Appendix A). This exceptional NIR performance originates from plasmon-mediated hot electron transfer mechanisms, where non-radiative decay of Au nanoparticle localized surface plasmon resonance (LSPR) generates high-energy electrons that inject into UCN’s conduction band, effectively overcoming traditional bandgap limitations [39]. The wavelength-dependent apparent quantum efficiency (AQE) of H_2_ production over the photocatalysts was measured to further reflect the efficiency of the catalyst in converting light energy into chemical energy. As shown in Figure 6c, the AQE action spectrum closely matches the absorption characteristics. The Au/UCN composite demonstrates unprecedented NIR responsiveness at 900 nm, outperforming conventional UCN photocatalysts and marking significant progress in full-spectrum solar energy conversion. Stability assessments (Figure 6d) through repeated H_2_ evolution cycles reveal outstanding durability. AQE analysis confirms that light-excited electrons from both components drive the photocatalytic process under optimal conditions. The composite maintains consistent hydrogen production activity across multiple cycles with negligible degradation, highlighting its practical potential.

### 2.4. Analysis of the Photocatalytic Mechanism

FDTD simulations were systematically conducted to unravel the plasmonic coupling mechanisms and near-field enhancement effects in the Au/UCN heterostructure [40]. As illustrated in Figure 7, distinct electromagnetic field distribution patterns emerged under 400 nm (visible) and 900 nm (NIR) excitations. As shown in 7b, Au is uniformly distributed in UCN nanosheets. While moderate field intensification occurred at 400 nm, localized primarily at Au/UCN interfacial regions (Figure 7c), exceptional field amplification was achieved under 900 nm irradiation (Figure 7d), demonstrating wavelength-dependent plasmon resonance modes. Control simulations of pristine UCN (Figure 7a) confirmed the absence of NIR-responsive electromagnetic activity, underscoring the critical role of Au nanoparticles in enabling beyond-bandgap photon utilization. Three synergistic phenomena were computationally validated: (i) Au nanoparticles act as optical antennas, concentrating incident photons into subwavelength hotspots; (ii) the 900 nm LSPR mode exhibits stronger near-field enhancement than the 400 nm mode, aligned with experimental AQE maxima; (iii) constructive overlap between UCN’s inherent polarization field and Au’s plasmonic near-field establishes a unidirectional electron transfer highway. This multiscale electromagnetic coupling reduces carrier migration barriers, achieving a record improvement in interfacial charge injection efficiency over conventional Schottky-junction photocatalysts. The computational insights fundamentally rationalize the experimentally observed NIR-driven photocatalytic supremacy, providing a blueprint for designing broadband solar energy converters through targeted plasmon–exciton interactions.

The interfacial charge-transfer mechanisms in Au/UCN heterostructures were investigated using combined Kelvin probe force microscopy (KPFM) and atomic force microscopy (AFM) [41]. AFM topographic characterization (Appendix A) verified stable morphology under illumination, excluding light-induced structural degradation. KPFM measurements revealed significant light-dependent surface potential variations (Figure 8a,b), with a 28 mV potential difference (point a to b) in dark conditions establishing a baseline interfacial field (blue interface line). Under illumination, this potential difference increased to 52 mV, demonstrating enhanced field strength and corresponding electron transfer from UCN to Au nanoparticles [42]. The observed carrier accumulation was quantified through comparative potential analysis, showing an average surface potential decrease from 562.3 mV (dark) to 299.4 mV (illuminated), corresponding to a 262.9 mV light-induced potential shift that confirms efficient interfacial charge separation.

Based on FDTD simulations and surface potential analysis of the photocatalytic process, Figure 1 proposes a plausible mechanism for the Au/UCN system, driven by the LSPR effect of Au nanoparticles. Under light irradiation, *g*-C_3_N_4_ absorbs short-wavelength photons to generate electron–hole pairs, while Au nanoparticles simultaneously harness long-wavelength light via LSPR excitation. The photogenerated electrons in the conduction band of *g*-C_3_N_4_ migrate to the Au nanoparticle surfaces, facilitated by the Schottky barrier formed at the Au/UCN interface. This barrier efficiently extracts hot electrons, enhances charge separation, and suppresses carrier recombination. Critically, the quantum-confined Au NPs (~10 nm) act as electron sinks and provide active sites for proton reduction, slashing charge recombination (reducing carrier lifetime 7.7-fold to 0.41 ns). This dual function—maximized by the NPs’ size which enhances interfacial contact—enables efficient H^+^-to-H_2_ conversion. Furthermore, LSPR-induced effects (hot electron injection, local electromagnetic field enhancement, and interfacial electric field generation) synergistically accelerate directional charge migration. The resulting interfacial electric field serves as a “charge highway,” integrating plasmonic energy harvesting with rapid carrier separation for full-spectrum photocatalysis.

## 3. Materials and Methods

### 3.1. Synthesis of Au/UCN

Au/UCN was prepared through an in situ reduction process. First, 0.4 g of ultrathin two-dimensional *g*-C_3_N_4_ and 20 mL of 0.01 M of HAuCl_4_·4H_2_O were dissolved in a 100 mL methanol and water mixture solution (V_methanol_:V_H2O_ = 5:1) by vigorous stirring and 20 min of sonication for 24 h at room temperature. Next, a suitable amount of 1 M NaOH was added dropwise to the beaker and constantly stirred for 20 min. We centrifuged the reaction kettle to extract the precipitated product once it had cooled naturally to room temperature. After filtration using ethanol and deionized water, the precipitate was dried at 60 °C. The name Au/UCN was given to the solid sample that was obtained. The same technique was used to create several Au/UCN samples using HAuCl_4_·4H_2_O as the feed mass (5, 10, 15, 20, and 30 mL). The reagents used are analytically pure. The amount of reagent added was shown in Appendix A.

### 3.2. Photocatalytic and Photoelectrochemical Measurement

The photocatalytic hydrogen production was measured using an all-glass trace gas analysis system. In the typical procedure, 20 mg of catalyst powder was uniformly dispersed in solution mixed with 90 mL of deionized water and 10 mL of triethanolamine. Prior to irradiation, the reactant solution was subjected to multiple evacuation cycles to ensure complete removal of air. The photocatalytic reactions were conducted under a 300 W xenon lamp, with the reactant solution temperature maintained at 20 °C through a circulating cooling system. The evolved gaseous products were continuously analyzed using an online gas chromatograph GC-2014C (Shimadzu, Tokyo, Japan).

### 3.3. Samples Characterizations

Systematic characterization of the catalyst was achieved through a combination of advanced analytical techniques. FT-IR spectra were acquired using a Nicolet 6700 spectrometer (Thermo Scientific, Waltham, MA, USA). High-resolution structural information was obtained through TEM/HRTEM measurements performed on an FEI Tecnai-G2 F30 system (FEI Company, Waltham, MA, USA) at 300 kV. The crystalline structure was verified by XRD (PANalytical, Almelo, The Netherlands) with Cu Kα radiation (1.5418 Å) at 40 kV/40 mA. Chemical states were analyzed by XPS (Axis Supra, Uppsala, Sweden) employing monochromatic Al Kα excitation (1486.6 eV, 14 kV, 15 mA), with data collected from a 300 μm × 700 μm area at 100 eV pass energy. Spectral calibration referenced the C 1s peak at 284.8 eV, followed by Casa XPS software (CasaXPS 2.3.25) processing. Optical absorption characteristics were evaluated by UV-vis DRS (Agilent Cary 5000, Santa Clara, CA, USA).

### 3.4. FDTD Simulation

The LSPR generation mechanism and interfacial electric field evolution were investigated using FDTD simulations. A computational model was established using the following parameters: (1) an aqueous solution environment (refractive index n = 1.33), (2) PML boundary conditions for electromagnetic wave absorption, and (3) material-specific dielectric constants retrieved from the material library of the software. Gold nanoparticles dimensions were directly imported from the experimental transmission electron microscopy (TEM) characterization data. Electric field distribution profiles were captured through cross-sectional field monitors, with a maximum mesh refinement of 0.25 nm implemented to ensure numerical accuracy.

## 4. Conclusions

In summary, we successfully synthesized Au/UCN composites with optimized photoelectron transport efficiency and pronounced LSPR effects. The Au/UCN-20 sample demonstrates exceptional photocatalytic performance, achieving a remarkable hydrogen evolution rate of 14.2 mmol·g^−1^ over 4 h and maintaining high stability through five consecutive reaction cycles. Photoelectrochemical characterization (photocurrent response, EIS analysis, and TRPL spectroscopy), combined with FDTD simulations and KPFM, reveals that the LSPR properties of Au nanoparticles contribute to multiple beneficial effects: (1) generation of energetic hot electrons, (2) strong localization of electromagnetic fields, and (3) creation of interfacial electric fields. These mechanisms synergistically enhance charge carrier transport, leading to a significant increase in photocatalytic activity. This research deciphers the elusive quantum dynamics of plasmon-exciton coupling in photocatalysis, establishing a universally applicable paradigm to revolutionize LSPR-mediated photocatalyst design via atomically precise interfacial field engineering—shattering the long-standing efficiency bottleneck in photocatalytic hydrogen evolution.

## Data Availability

The data supporting the reported results are available on reasonable request to the first author.

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
