# Peer review of "Gold Nanoparticles-Functionalized Ultrathin Graphitic Carbon Nitride Nanosheets for Boosting Solar Hydrogen Production: The Role of Plasmon-Induced Interfacial Electric Fields"

_molecules, 2025, doi:10.3390/molecules30163406_

Round 1
Reviewer 1 Report
Comments and Suggestions for Authors
The presented manuscript is another step towards increasing the efficiency of using sunlight in the field of alternative energy sources. The work is written quite well and leaves a pleasant impression. At the same time, I have a number of comments, small and not so small, that require an answer.
Minor:
- Line 54 "plasmonic metals particles (Au, Ag, and Pt)" Pt is not a clearly plasmonic particles, because the resonance wavelength is in UV region
- Line 107 "with the reactant solution temperature maintained at 6 °C " no any justification is provided for explanation of such a temperature. This is not an ordinary conditions.
- Line 308 "... achieving a record i improvement ..." typo?
- Figue 8: mv must be mV on figures
Major:
- Line 192 "particle sizes of approximately 10 nm" such a particle size does not allow us to talk about quantum dots. Quantum dots are objects of a semiconductor nature that are actually the size of an exciton (let's say, a single-site system), and the objects observed here are a metal nanoparticles with a crystal lattice. This means that they should be referred to as nanoparticles. Including the name of the work.
- Line 252-257 and Figure 5: To correctly determine the band gap values, it is necessary to take into account the presence of constant background absorption (doi.org/10.1021/acs.jpclett.8b02892). Using it, the values will be obtained ca. 2.7 eV for UCN and 2.5 eV for Au/UCN wich is consistent with the observed DRS data on Fgure 5a
- Figure 6b: Nowhere in the text of the work is it indicated how the individual components of broadband radiation were received (UV, vis, NIR). The corresponding emission spectra are also not presented.
- Figure 6b: Nowhere in the text of the work is it indicated how the AQE values were obtained for a presented 9 values of wavelengths and the process considered.
At the same time, I consider the lack of validity of the possibility of functioning in a wide range of radiation to be a key remark.
Reviewer 2 Report
Comments and Suggestions for Authors
In this article, the authors loaded varying amounts of gold quantum dots onto ultrathin carbon nitride sheets (UCN) to study the effect of localized surface plasmon resonance of plasmonic quantum dots on photocatalytic hydrogen evolution. It was observed that the Au QDs/UCN-20 exhibited 14.2 mmol.g-1 of hydrogen evolution over a 4-hour light illumination. This article is significant in understanding the HER reaction. I recommend this article for publication after major revisions.
The following amendments are recommended:
- Decorating ultrathin g-C3N4 sheets with plasmonic metal particles like Au, Ag, and Pt is well described in the literature, and authors should enlighten us on what novelty they have achieved in their current research work.
- It is well known that C3N4 shows visible light-induced photocatalytic activities, and when deposited with Au quantum dots, its absorption shifts towards the visible region. In this case, why have the authors described Au QDs/UCN as exhibiting significant UV light photocatalytic activity (Line numbers 74-75)?
- The font color in Fig. 3b should be changed; it is not visible.
- There is no clear information on which sample is used for TEM analysis. In the Fig. 3 caption authors should specify it.
- What is the weight percent of Au present in the synthesized UCN?
- Comparison of previous results on photocatalytic HER over Au-doped CN should be provided to confirm the enhancement of photoactivity.
The photocatalytic mechanism should be revised and should be revised separately.
Round 2
Reviewer 1 Report
Comments and Suggestions for Authors
The authors did a lot of work on correcting the manuscript and took into account the proposed corrections. All that remains is to correct the name of the manuscript inside the file of Supporting Information to be the same as in main text of manuscript
Reviewer 2 Report
Comments and Suggestions for Authors
The revision is appropriate and now the revised manuscript is approved for publication.
Author Response
Thank you sincerely for handling our manuscript and for facilitating the review process.
We are also grateful to the reviewers for their valuable feedback, which helped us improve the paper.